# Milk Fat Globule Epidermal Growth Factor VIII Fragment Medin in Age-Associated Arterial Adverse Remodeling and Arterial Disease

**DOI:** 10.3390/cells12020253

**Published:** 2023-01-07

**Authors:** Mingyi Wang, Kimberly R. McGraw, Robert E. Monticone

**Affiliations:** Laboratory of Cardiovascular Science, Intramural Research Program, National Institute on Aging, National Institutes of Health, Biomedical Research Center, 251 Bayview Boulevard, Baltimore, MD 21224, USA

**Keywords:** medin, degenerative inflammation, cardiovascular remodeling, cardiovascular disease, cerebrovascular cognitive decline and dementia

## Abstract

Medin, a small 50-amino acid peptide, is an internal cleaved product from the second discoidin domain of milk fat globule epidermal growth factor VIII (MFG-E8) protein. Medin has been reported as the most common amylogenic protein in the upper part of the arterial system, including aortic, temporal, and cerebral arterial walls in the elderly. Medin has a high affinity to elastic fibers and is closely associated with arterial degenerative inflammation, elastic fiber fragmentation, calcification, and amyloidosis. In vitro, treating with the medin peptide promotes the inflammatory phenotypic shift of both endothelial cells and vascular smooth muscle cells. In vitro, ex vivo, and in vivo studies demonstrate that medin enhances the abundance of reactive oxygen species and reactive nitrogen species produced by both endothelial cells and vascular smooth muscle cells and promotes vascular endothelial dysfunction and arterial stiffening. Immunostaining and immunoblotting analyses of human samples indicate that the levels of medin are increased in the pathogenesis of aortic aneurysm/dissection, temporal arteritis, and cerebrovascular dementia. Thus, medin peptide could be targeted as a biomarker diagnostic tool or as a potential molecular approach to curbing the arterial degenerative inflammatory remodeling that accompanies aging and disease.

## 1. Introduction

Aging exponentially increases the prevalence of hypertension, aneurysm/dissection, vascular-related cognition declines and related dementia due mainly to adverse structural and functional vascular remodeling [1,2,3,4,5,6]. Age-associated adverse vascular remodeling, including inflammation, elastolysis, fibrosis, calcification, and amyloidosis, is the histopathologic foundation for the initiation and progression of age-associated vascular endothelial dysfunction and stiffening [1,2,3,4,5,6,7,8]. The age-associated vascular cellular and extracellular inflammation drives endothelial layer destruction/dysfunction and vascular stiffening, and eventually facilitates the initiation and progression of age-associated vascular disease such as hypertension, aneurysm/dissection, vascular-related cognitive decline, and related dementia [2,8,9].

A glycoprotein, milk fat globule epidermal growth factor VIII (MFG-E8), also known as lactadherin, is mainly secreted by vascular smooth muscle cells (VSMCs) in the arterial wall [10]. In humans, MFG-E8 is made up of the 364-amino acids, including an epidermal growth factor (EGF) domain in the N-terminus and two discoidin-like domains in the C-terminus (Figure 1) [11]. Medin is a 50-amino acid peptide, enzymatically cleaved from the second discoidin-like domain from the residue 245 to 294 of MFG-E8 and are the most common amyloidogenic in the vascular wall in subjects aged over 50 years (Figure 1) [10,12,13,14,15,16,17]. Notably, MFG-E8 is an abundant protein, which is markedly increased in the aortic wall in mice, rats, nonhuman primates, and human samples with aging (Figure 2) [7,18,19,20,21]. MFG-E8 connects VSMCs to elastic fibers via the EGF domain and medin region in the elastin-contractile unit in the vascular wall (Figure 3).

Medin amyloid is a common amyloid protein (~97%) observed in the media of aortic walls of humans above 50 years of age [12,14,15,16,22,23,24]. Both the signaling of medin and its parent molecule MFG-E8 are increased in the aged arterial wall, promoting inflammatory vascular remodeling in hypertension, diabetic vasculopathy, aneurysm/dissection, arteritis, and cerebrovascular related dementia [1,2,4,7,12,14,16,17,20,21,22,25,26,27]. In vitro, ex vivo, and in vivo investigations reveal that medin promotes inflammatory phenotypic shifts of both endothelial cells (ECs) and VSMCs, contributing to age-related endothelial dysfunction and arterial stiffening, a fertile soil for the initiation and progression of aortic aneurysm, dissection, and vascular-related dementia (Figure 4 and Figure 5) [2,3,4,21,26,27,28,29,30].

The inflammatory role of MFG-E8 in vascular remodeling and vascular disease with aging have been previously examined [10,26]. Our current review mainly focuses on how (1) aging increases MFG-E8 and its fragment medin during inflammatory vascular remodeling; (2) medin drives the inflammatory phenotypic shifts of ECs and VSMCs; (3) medin mediated vascular cellular inflammatory reactions modify the vascular extracellular matrix substrates such as amyloidosis, elastic fragmentation, and calcification; (4) medin inflammatory signaling and modifications promote endothelial dysfunction and arterial stiffening; and (5) medin mediated vascular cellular and extracellular remodeling is a fertile ground for the development of the vascular diseases such as aortic aneurysm/dissection, temporal arteritis, vascular related cognitive impairment and dementia with advancing age (Figure 4 and Figure 5).

## 2. MFG-E8 and Its Fragment Medin Expression during Age-Associated Inflammatory Remodeling

Aging increases MFG-E8 expression in the arterial wall in mice, rats, nonhuman primates and human samples; and aging also increases MFG-E8 fragment medin in the aortic wall in mice and human samples [1,4,7,12,16,17,18,19,21,28]. The levels of MFG-E8 abundance are closely associated with age-associated intimal medial thickening, elastolysis, calcification, angiotensin II (Ang II)/ its receptor AT1 protein signaling and nuclear transcriptional factor-kappa B (NF-κB) activation in mice; in contrast, MFG-E8 deficiency markedly alleviates these age effects in mice [18,19,31]. It is known that Ang II signaling is a driving force for arterial aging [3,19].

Ang II is abundantly colocalized with MFG-E8 in the aged arterial wall in rats and human samples [21]. Treating both young and old rat VSMCs with Ang II increases MFG-E8 expression in a dose-dependent manner [21]. Furthermore, a chronic infusion of Ang II or saline control via an osmotic minipump to both young adult MFG-E8 knockout (MFG-E8 KO) and age-matched wild type (WT) mice for 28 days demonstrates that Ang II markedly enhances aortic MFG-E8 expression in WT animals; however, MFG-E8 was not detected in MFG-E8 KO animals [19]. The above findings suggest that MFG-E8 is a downstream molecule of Ang II signaling in the arterial walls or VSMCs in rats or mice. Further study shows that MFG-E8 mediates Ang II induced vascular inflammatory remodeling such as intimal medial thickening, collagen deposition, elastic fiber fragmentation in the arterial wall in young mice mimicking that of old untreated mice, independent of blood pressure increase [19]. Thus, these findings suggest that MFG-E8 is an important mediator of Ang II-induced arterial aging in mice.

Aging exaggerates neointima formation and inflammation in the ligated common carotid artery in mice, possibly due to increases in MFG-E8 abundance and leukocyte infiltration [18]. Interestingly, a local treatment with a recombinant MFG-E8 further enhances leukocyte infiltration, NF-κB transcription activation, intercellular molecular adhesion 1(ICAM1), and vascular cellular adhesion molecule 1 (VCAM1) expression, and the vascular cell proliferation in the injured common carotid artery with aging [18]. These findings suggest that MFG-E8 promotes and accelerates arterial inflammation with advancing age.

Remarkably, growing evidence indicates that the amyloidogenic MFG-E8 fragment medin also exerts an inflammatory role in the arterial wall; however, what, when, and how local proteinases cleave MFG-E8 into the intracellular or extracellular medin in the arterial wall during aging remains enigmatic. For example, abundant proteinases, such as matrix metalloproteinase type II (MMP-2), are markedly activated in the arterial wall with aging; however, it is unknown how the activated proteinase cleaves MFG-E8 into medin and promotes medin aggregation within the aged arterial wall [3,26]. Thus, it is warranted to elucidate the underlying cellular and molecular mechanisms behind enzymatically cleaving MFG-E8 into medin for the diagnosis and treatment of amyloid fibril formation and amyloidogenic inflammation in the arterial wall with advancing age.

## 3. Medin Induces Vascular Cell Inflammation

Medin is positioned in a microenvironment enriched with an increase of superoxide while there is a decrease of bioavailable nitric oxide in vascular walls in both humans and mice [7,28,32]. This metabolite stressful niche alters the aggregation process of medin, per se, and facilitates potential posttranslational modifications such as oxidation and nitration [32]. Medin could be nitrated or oxidized at tyrosine and tryptophan residues, with resultant effects on morphology that lead to longer fibrils with increased cytotoxicity such as senescence, apoptosis, and necrosis to either human ECs or human VSMCs [2,7,23,28,32,33]. Medin-associated oxidative stress and inflammatory reactions induce vascular cell inflammatory phenotypic shifts and vascular dysfunction (Figure 4).

### 3.1. EC Inflammation and Viability

Maintaining the integrity of the endothelial layer is key to vascular health by preventing vascular wall inflammation, thrombosis, embolus, bleeding, platelet activation, attachment, and aggregation [3,6]. However, both aging and metabolic disorders such as hypertension and obesity increase the susceptibility of ECs to senescence, apoptosis, and/or necrosis due to increases in inflammation and oxidative stress [3,6]. Medin or its parent molecule MFG-E8 enhances inflammation and oxidative stress in human ECs via advanced glycation end-products (AGEs)/receptor AGEs (RAGE) signaling [28,34,35].

Treatment with medin peptide increases reactive oxygen species (ROS)/reactive nitrogen species (RNS) production, and apoptosis reduces nitric oxide (NO) production, proliferation, the migration rate, and the viability of human ECs [2,28]. In addition, medin treatment increases the gene and protein expression of interleukin-6 (IL-6) and interleukin-8 (IL-8) through the activation of transcription factor NF-ĸB in human ECs; however, these effects are substantially reversed by the antioxidant, polyethylene glycol superoxide dismutase, a nuclear factor erythroid 2-related factor (Nrf2) activator or by the RAGE inhibitor FPS-ZM1 [2,28]. Notably, Nrf2 activation increases antioxidant defenses such as increases in heme oxygenase 1 (HO-1), NAD(P)H quinone dehydrogenase 1 (NQO1), and superoxide dismutase 1 (SOD) in human ECs [2].

Overexpressing MFG-E8 induces EC apoptosis via the downregulation of B-cell lymphoma protein 2 (Bcl-2) and its-associated X protein (Bax), the release of cytochrome *c*, and the activation of caspase-9 and caspase-3; however, downregulation of MFG-E8 expression reduces caspase-3 activity, inhibits the AGEs-induced EC apoptosis, and subsequently lowers inflammation and maintains endothelial health [34,35].

Remarkably, it is an emerging concept that impaired autophagy, i.e., a decrease of the ratio of microtubule-associated protein 1A/1B-light chain 3 III (LC3 II/I), has been implicated in human endothelial survivability with aging [33]. Interestingly, human umbilical cord ECs exposed to medin for 20 h substantially downregulates LC3 II/I relative levels, impaired autophagy, and subsequently reduces EC viability, further supporting this concept [33].

Notably, changes in the aortic amyloid polypeptide medin aggregation incubated with heparin alters the morphology of the amyloid fibrils and eliminates the small cytotoxicity of human ECs, forming benign fibrils [23]. Small peptide β-alanine inhibitors disrupts hydrogen bonding and prevents the aggregation of medin, which also reduces the toxicity of human ECs [23]. In addition, the C-terminal domain of pro-lung surfactant protein C can effectively prevent fibril formation in medin [36]. These findings suggest that the management of medin amyloid aggregation could be an approach to treating amyloid disease.

### 3.2. VSMC Inflammation and Viability

VSMCs are the predominant cell type in the arterial wall. In the normal healthy adult, VSMCs display a contractile phenotype to help maintain vascular tone and blood pressure [3]. Under metabolic stressful conditions such as aging, hypertension, and metabolic syndrome, VSMCs phenotypes are switched from a contractile to a synthetic state [3,6]. Indeed, aging increases the number of senescent VSMCs, which release a large amount of extracellular vesicles such as exosomes containing both MFG-E8 and medin, and accelerates medin aggregation and amyloid fibril formation in the arterial wall [37].

Medin amyloid has been found to be deposited between VSMCs [1,12,13,23,24]. The accumulation of aggregated medin is toxic to the surrounding VSMCs and promotes the secretion and activation of inflammatory matrix metalloproteinase type II (MMP-2), contributing to cellular necrosis [24]. Medin also induces oxidative species, increasing superoxide and decreasing bioavailable nitric oxide in ECs, which eventually contributes to the toxicity of VSMCs in a paracrine-dependent manner [32,38]. Conversely, oxidative stress accelerates the aggregation of medin and nitrates on the tyrosine and tryptophan residues of medin, resulting in increased fibril length, which leads to the further cytotoxicity and decreased viability of VSMCs [32,38].

Interestingly, the medin parent molecule MFG-E8, is released in exosomes from human ECs treated with high glucose, which induces the senescence of human VSMC in a paracrine manner [39]. Conversely, senescent VSMCs secrete exosomes that promote the secretion of MFG-E8 and accelerates the aggregation of medin in the arterial wall [37]. Notably, these senescent VSMCs become osteogenic cells, and are characterized by increases in the expression of MFG-E8, alkaline phosphatase (ALP), runt-related transcription factor 2 (Runx2), and calcification [39].

In addition, medin oligomers form membrane pores, which induce unregulated ionic currents, including Mg^2+^, Ca^2+^, and CI^−^. These membrane pores can alter cellular homeostasis, leading to the necrosis of VSMCs [40]. In fact, it has been reported that medin amyloid pores perturb the membrane and permeability of VSMCs, releasing necrosomes and subsequently leading to necroptosis [40].

## 4. Medin Remodels Extracellular Matrices

Medin and its parent MFG-E8 are not inert in the extracellular matrix, but rather are potent molecular cues, actively participating in extracellular matrix remodeling such as amyloidosis, elastic fiber degeneration, and calcification via an increase of the inflammation of VSMCs (Figure 5).

### 4.1. Amyloidosis

Arterial amyloidosis is the aggregation of amyloidogenic proteins in the arterial wall. So far, up to 30 amyloid proteins or peptides, including medin, have been found in the human body and are characterized by a β sheet secondary structure, a fibrillar morphology diameter of 7–13 nm, and appear red when stained with Congo red, and apple-green birefringence when observed under a polarized microscope [15,41].

Medin is the most common form of amyloidogenic polypeptide observed in old aortic and cerebrovascular walls in both murine animals and humans [1,2,4,12,13,14,15,16,17,22,27,28,32,33,41]. Murine medin shares 78% of amino acid sequences with human medin [1]. Both medin and its parent molecule MFG-E8 are abundantly detected together in the amyloid aortic and cerebrovascular walls [1,27,42]. Interestingly, both medin and MFG-E8 are extracellularly aggregated in an age-dependent manner in the arterial wall [1,7,29,43].

Amyloid medin fibril is formed through an aggregation-prone region in the last 18–19 amino acid residues and the C-terminal phenylalanine, favoring amyloid fiber formation [15]. An ex vivo study demonstrates that medin amyloid fibrils could be generated from medin and medin fragments in the indicated time of incubation in Table 1 [15].

In the arterial wall, medin can present in native, misfolded, oligomers, protofibrils, or amyloid fibrils, which may be associated with the different phases of in vitro amyloid medin dynamic fibril formation, including lag, growth, and the plateau or saturation phases (Figure 1) [15,24,40,43,44,45].

Aggregated, thinly streaked medin, as observed under the microscope, is detected along with elastic fibers within the aortic wall and along with the internal elastic lamina (IEL) in the temporal artery and intracranial vessels; and sporadic aggregated nodule-like medin has also been observed in old aortic walls in both mice and humans [1,11,12,13,14,15,16,17,24,36,43,44,45,46,47].

Medin amyloid fragments commonly interact with classic amyloidogenic proteins AA and Aβ, facilitating the aggregation of these amylogenic proteins into fibrils by a cross-seeding mechanism ex vivo and in vivo [13,27]. Remarkably, the amino acid sequence of medin strikingly resembles the sequence of that of Alzheimer’s disease (AD) amyloid β (Aβ) polypeptides around the structural turn region [38]. Medin shares 16% of its global sequence with the Aβ peptide [12]. Medin and Aβ have high local similarities with 23–39 residues [12]. Importantly, medin amyloid promotes the initiation and propagation of Aβ in the arterial wall [13]. In seven aortic walls from patients with diagnosed systemic serum amyloid derived A (AA) amyloidosis, four displayed partial co-localization between medin and AA aggregates (~57%) [13]. Thus, the cross-seeding of fibrilization is the main molecular mechanism by which common medin amyloidosis potentially initiate the formation of the more uncommon amyloidosis in the older arterial wall.

In addition, both MFG-E8 and medin interacts with degenerated elastic fibers, suggesting that elastin-derived fragments are probably important elements in the formation of MFG-E8/medin-associated amyloidosis, and significantly affect the elasticity of elastic fibers [14,22,48,49]. Interestingly, medin-induced negative charged desmosomes in the elastic fragments and negative charged phospholipid membranes derived from apoptotic ECs and VSMCs likely promotes medin self-aggregation and amyloid formation via an alpha (α)-helical intermediate and beta (β)-sheet confirmation [46].

### 4.2. Elastolysis

Elastic fibers are composed of a polymerized tropoelastin, the core elastin protein, and are surrounded by fibril-rich microfibrils [5]. Elastic fibers play an important role in the resilience and elasticity of the arterial wall, the sequestration of transforming growth factor-beta 1 (TGF-β1), and the regulation of TGF-β1 activation [5]. MFG-E8 provides a direct connection between VSMCs with the RGD motif on the EGF-like domain and elastic fibers via the medin binding domain and has a structural role in contractile-elastin unit remodeling in the large arterial wall with advancing age (Figure 3) [11,12,14].

Medin contains the elastin-binding domain of MFG-E8 [12,14]. Elastin is a very hydrophobic molecule, and it is possible that when MFG-E8 comes in close contact with elastin, it displays the hydrophobic patch of medin, which then interacts with the elastin, forming an amyloid compound [44]. The displayed MFG-E8/medin moiety may then be cleaved and attract more, which could initiate the formation of medin amyloid fibrils along with elastic fibers in a feed forward manner [1,14,24]. In fact, the medin amyloid is frequently observed along with the fragmented elastic laminae, supporting the theory that degrading elastin-derived peptides may be important for the initiation and elongation of medin amyloid fibrils [1,14,22,27,48].

Medin is clearly associated with elastic fibers observed under an electron microscope [14,22,27]. Immuno-electronic microscopy has further proved that medin amyloid deposits appear topographically to be very closely associated with fractured elastic fibers [14,17,22,27]. An in vitro study indicates that medin binds to tropoelastin in a concentration-dependent fashion [14]. Importantly, treating VSMCs with medin induced the secretion of activated MMP-2 [22,24]. In turn, the activated MMP-2 interacted with the tropoelastin of the elastic fibers and effectively degraded tropoelastin and subsequently cleaved elastic fibers [5,19,31]. Notably, the compounds of fragmented elastic fragments immunolabelled for medin are often found to be in the temporal inflammatory arterial wall and are engulfed by phagocytes such as macrophages, which are the main cellular and molecular mechanism of the fragmented elastic fibers in the temporal arteritis [17].

### 4.3. Calcification

Aging increases ectopic calcification, the deposition of hydroxyapatite crystals in the vascular wall [3,6]. Medin and its parent molecule MFG-E8 have been reported to be involved in the initiation and progression of arterial calcification [39,42,50,51,52,53]. The MFG-E8 protein includes EGF-like and discoidin/coagulation factor 5/8 (F5/8C) domains (Figure 1). The three-dimensional structures of MFG-E8 suggest that these domains have a propensity to bind calcium and extracellular vesicles such as exosomes [53]. MFG-E8 chromosomal locations reveal a nested syntenous relationship with two calcification genes: hyaluronan and proteoglycan link protein 1/3 or cartilage link protein (hapln1/hapln3) and verscan or chondroitin sulfate proteoglycan (vcan/can) on the same chromosome in several vertebrae classes, including humans [53]. In fact, previous findings have demonstrated that MGE-E8 is a potent calcium binding protein and an element of tissue calcification or biomineralization [51,52,53,54]. For example, MFG-E8 has been detected in calcified eggshells [51,52,53]. In addition, the MFG-E8 protein is associated with extracellular vesicles containing amorphous calcium carbonate that is a typical mineralization site [42,50,53]. Importantly, MFG-E8 deficiency significantly alleviates elastic fiber degradation, fibrosis, and calcification in the arterial walls in MFG-E8 knockout mice compared with WT animals with advancing age [31]. Furthermore, the absence of MFG-E8 markedly downregulates the calcification molecule runt-related transcription factor 2 (RUNX2) and ALP expression in aging MFG-E8 knockout mice versus wild-type animals [31]. In addition, treating VSMCs with MFG-E8 markedly increases the levels of RUNX2 and ALP calcification molecules. A recent study indicates that MFG-E8 promotes calcification and osteogenetic trans-differentiation of VSMCs in common carotid arteries after ligation by upregulating TGF-β1/BMP2, SMAD 2/3, PAI-1, MMP-2, collagen, and Runx2 in mice in vivo and in vitro [50]. These in vitro, in vivo, and genetic studies strongly suggest that MFG-E8 plays a pivotal role in inflammatory calcification in the aging aorta or injured carotid arterial wall.

Interestingly, MFG-E8 secreted from senescent vascular endothelial and smooth muscle cells collectively facilitate vascular inflammatory calcification [39,42]. Aging increases the prevalence of metabolic syndrome, including hyperlipidemia. Hyperlipidemia, an inflammatory condition, stimulates the secretion of exosomes, 20–100 nm vesicles, from ECs, upregulates the expression of MFG-E8, and induces the inflammatory mediator secretions such as IL-1β, IL-6, and IL-8 via a downregulation of sirtuin1 (SIRT1) [39]. Increased exosomal MFG-E8 from ECs promotes VSMC transdifferentiating into a senescent calcification state, characterized by enhanced expressions of biomineralized molecules ALP and Runx2, contributing to an increase of the mineralized impetus of extracellular nodules for the initiation and progression of calcification in a paracrine-dependent manner [39].

Notably, MFG-E8′s fragment medin has three amyloidogenic regions, forming calcium-based metal-organic fragments, and facilitates the initiation and propagation of ectopic calcification in the arterial wall [42]. Medin treatment increased the release of exosomes from ECs in the arterial wall [42]. Medin treatment also induces osteogenic trans-differentiation of VSMCs and the secretion of the macrovesicles, exosomes, and apoptotic bodies from these vascular cells, promoting the formation of the calcifying nidus that is involved in the initiation and progression/propagation of calcified/mineralized matrix [42].

## 5. Medin Deposition Impairs Vascular Function

The remodeled or modified vascular ECM and inflammatory vascular cells contribute to the dysfunction of the vascular wall. The internal elastin laminae have a high affinity to medin peptide, forming amyloids [14,17,22]. The medin-associated internal elastic lamina greatly impacts on the microenvironment of the overlying endothelial layer, promoting the structural and functional deterioration, likely contributing to endothelial dysfunction in larger and smaller arteries (Figure 4 and Figure 5). Amyloid internal elastic fibers are potentially susceptible to fracture and the loss of elasticity; and this stiffened physical barrier is harmful to the survival of ECs and susceptible to the medial invasion of VSMCs, likely leading to VSMC infiltration of the subendothelial space; and this stiffened physical barrier also changes its chemical property, such as with the increased accumulation of oxidized low-density lipoproteins in fractured internal elastic ends [3]. All these alterations may contribute to the stiffening of the vascular wall (Figure 4 and Figure 5).

### 5.1. Arterial Endothelial Disorder and Dysfunction

Aging increases endothelial disorder and dysfunction in the arterial wall. Endothelial-dependent vasodilation declines with advancing age in both men and women [3]. Aging impairs the endothelial physical layer via endothelial apoptosis and senescence and increases the endothelial cellular gap due to alterations of endothelial cellular junctions, facilitating increases in inflammation, thrombosis, and permeability [3]. In large arteries, endothelial dysfunction also promotes intimal medial thickening, arterial stiffening, and increases in blood pressure [3]. In the small arteries such as in the brain, aging damages the cerebrovascular endothelial layer, promoting inflammation, thrombosis, permeability, micro-bleeding/thrombosis/embolism, and decreases the blood perfusion, leading to an insufficient supply of nutrients and oxygen and producing reactive oxidized/nitrative species (ROS/RNS), contributing to neurovascular unit structural and functional damage [1,2,4,13,27,28]. Medin treatment significantly damaged the endothelial-dependent dilation of human leptomeningeal arterioles [9,28]. Importantly, genetic deficiency of the medin precursor protein, MFG-E8, eliminates not only vascular medin aggregates but also prevents the age-associated decline of cerebrovascular endothelial dependent dysfunction [1]. Accumulating evidence indicates that medin is involved in the detrimental effects on arterial endothelial-dependent dilation and exerts an important role in the development of endothelial dysfunction with aging through increased EC apoptosis, oxidative stress, inflammation and or the decrease in NO bioavailability as well as the EC survival ability [2,22,28,33,46].

### 5.2. Arterial Microstructure Disorder and Stiffening

The pulse wave velocity (PWV), a clinical gold standard measure of arterial stiffness, increases with age in both men and women [3]. Arterial stiffening could be attributed to the increases in both vascular cells and matrix stiffness with advancing age. Atomic force microscopy (AFM) observations indicate that the stiffness of both elastic laminae and the inter-space of elastic laminae are increased with aging [55,56,57]. Furthermore, in vitro studies demonstrate that both ECs and VSMCs isolated from old animals become stiffer than those from young animals [58,59,60,61,62].

An oscillatory nanoindentation measurement can localize and determine the local vascular wall mechanical properties via the measurement of shear storage modulus, *G*’ and shear loss modulus, *G*˝ [22]. This examination using this nanoindentation technique shows that significantly lower *G’*, an index of reduction of elasticity of the tissue, has been detected in the elastin-contractile units, enriched with MFG-E8, medin, oligomer medin and medin amyloid fibrils [22]. In other words, the higher MFG-E8, oligomer medin, medin amyloid fibrils in the elastin-contractile units in the arterial wall convey higher levels of local vascular microstructure stiffness (Figure 3) [22]. Thus, an increase in medin amyloid greatly increases vascular wall stiffness, contributing to increasing blood pressure, aneurysm/dissection, cerebrovascular dysfunction, and cognition decline in the elderly.

## 6. Medin Deposition Is Involved in the Pathogenesis of Vascular Diseases

Medin has been reported to be elevated in aortic aneurysm/dissection, temporal arteritis, and vascular dementia [2,4,17,24]. Medin-associated vascular cell apoptosis and necrosis; medin-associated elastin degeneration, calcification, and amyloidosis result in a weakness of the vascular wall. Importantly, medin-associated endothelial dysfunction and arterial wall stiffening results in an imbalance of mechanical force on the local arterial wall. This medin-associated structural and function remodeling contributes to the pathogenesis of arterial aneurysm and dissection (Figure 5). The internal elastin laminae bound to the medin amyloid, trigger endothelial immune reactivation, leading to the dysfunction of the arterioles in the brain. Importantly, endothelial dysfunction causes an insufficient supply of blood to brain tissue by cerebrovascular tissue, leading to the brain ischemia and abnormal immune activation, eventually contributing to the pathogenesis of vascular-related dementia (Figure 5). In addition, medin and the associated fragment elastin peptide are engulfed by macrophages, causing giant cell vasculitis (Figure 5).

### 6.1. Arterial Aneurysms and Dissections

Arterial aneurysm and dissections are usually a complication of advanced stage hypertension, atherosclerosis, and diabetes [63]. Arterial aneurysm is an excessive focal enlargement/expansion of the artery caused by the weakening of the arterial wall. Arterial dissection is an abnormal, and usually abrupt, formation of a tear (intimal) along the inside wall of an artery [63]. The pathologic foundations in the arterial wall aneurysm/dissection are closely associated with the degradation of the intima and media, which includes EC, VSMCs, collagen, elastin, and elastin-contractile units [63]. Aneurysm and dissection areas become inflamed, stiffened, and fragile, which are closely associated to a disorder of cellular proteostasis and the quality of protein [63].

Interestingly, MFG-E8, its fragments and medin-derived amyloids, have been detected in patients afflicted by either thoracic aortic aneurysms or type A aortic dissection [22,24,29]. The abundance of MFG-E8, which produces medin, is elevated in the aortic media of older-aged subjects where amyloidosis is enhanced [1,24,40]. Previous findings suggest that MFG-E8 seems to be decreased in the aortic aneurysm or dissection while medin and medin amyloid are increased [11,22,24,63]. Remarkably, the number of senescent VSMCs is markedly increased in the aortic aneurysm/dissection which secrete extracellular vesicles, containing MFG-E8 and medin, promoting medin amyloid formation [37,64]. It has been reported that medin aggregates into amyloid, leading to potentially fatal conditions of thoracic aortic aneurysm and dissection [22,24,29]. Notably, medin is increased along with the biomarkers of oxidative stress 8-hydroxy-2’-deoxyguanosine and 4-hydroxy-2-nonenal in the aortic media of middle-aged or older-aged donors [7]. Oxidative damage induces the disruption of VSMCs, resulting in the decrease of α-actin, a highly expressed protein in contractile VSMCs, and matrix remodeling, causing the vulnerability of the aging arterial wall [30,32]. In addition, medin was the conspicuous trigger for necrosis in VSMCs and for the release of activated MMP-2, thus further promoting the degradation of the aortic matrix, facilitating the development of aneurysm/dissection [22,24]. The senescence and necrosis of VSMCs and the breakdown of elastic fiber networks is the pathologic foundation of the development of aortic aneurysm or dissection. These findings suggest that targeting MFG-E8 or its fragment medin is a novel molecular approach to diagnose, prevent, or treat aortic aneurysm/dissection.

### 6.2. Temporal Arteritis

Temporal arteritis, also known as giant cell arteritis, is an inflammatory disease of the blood vessels near the temple [65,66]. The prevalence of temporal arteritis ranges from approximately 0.5 to 27 cases per 100,000 people, aged 50 years or older [66,67]. In the clinic, temporal arteritis can cause damage to eyesight, including sudden blindness in one or both eyes, or damage other arterioles, leading to aneurysm/dissection, stroke, or a transient ischemic attack [65].

Medin amyloid deposits commonly occur along the internal elastic lamina of the temporal artery in the elderly [16,17]. In temporal artery biopsies from 22 patients with clinical and histological signs of giant cell arteritis, medin amyloid deposits were found in 14 (64%) biopsies [17]. Furthermore, it is interesting that, under a microscope, the medin amyloid appeared topographically closely related to the elastin-derived fragments [12,17]. Notably, fragmented elastic material was often immunolabelled for medin and found to be engulfed by giant cells known as macrophages [12,17]. Furthermore, in situ hybridization showed that MFG-E8 was expressed locally by VSMCs of the temporal artery [12,17]. These findings suggest that MFG-E8 or its fragment medin play a pathologic role in the inflammatory process in giant cell arteritis. Thus, targeting medin or MFG-E8 is a novel potential molecular approach to the diagnosis, prevention and treatment of temporal arteritis.

### 6.3. Vascular-Related Cognitive Impairment and Dementia

The endothelial dysfunction of the cerebrovascular system plays an important role in age-associated vascular contributions to cognitive impairment and dementia (VCID [2,9,68,69,70]). VCID is a decline in thinking capacity caused by conditions that block or reduce cerebrovascular blood flow or perfusion to various regions of the brain, depriving them of oxygen and nutrients [9,68,69,70]. Insufficient blood flow, ischemia, can damage and kill cells in the tissue, especially in the vulnerable brain in the elderly, and eventually alter their thinking skills [68,70,71]. Thinking defects can start as mild ischemia that gradually worsen because of multiple minor strokes, microbleeds or micro-embolisms that affect smaller blood vessels, promoting the widespread damage of a vascular-neuro unit and causing VCID [72,73]. The prevalence of VCID exponentially increases with advancing age in modern society. Notably, VCID is ranked as the second most frequent cause of dementia, exceeded only by Alzheimer’s disease (AD).

Vascular and cellular senescence is closely associated with cognition decline with aging [74,75]. VSMC senescence accelerates medin aggregation through small extracellular vesicle secretion and extracellular matrix reorganization [37,42]. A growing body of evidence documents that amyloid deposition is closely associated with cerebrovascular medin abundance in humans; and the medin amyloid load is higher in vascular dementia patients than in cognitively normal patients [2,4,9,28]. Medin amyloid is a key element of the neurovascular unit in the brain, causing microvascular endothelial dysfunction through oxidative and nitrative stress, and promotes inflammatory signaling in EC via RAGE, showing EC immune activation and neuroinflammation [2,9]. In addition, medin induces EC immune activation via the release of interleukin-6/-8 (IL-6/8), which subsequently modulates astrocyte inflammatory activation [2,28]. These immune reactions and inflammation promote the development of degenerative aneurysm and microstructure weakening [1]. This localized structure and functional inflammatory remodeling contributes to local ischemia in the brain and eventually causes cognitive decline [1]. Importantly, the medin-induced endothelial dysfunction and oxidative stress are markedly reversed by the antioxidant polyethylene glycol superoxide dismutase or by the RAGE inhibitor, FPS-ZM1 [2,28,33]. A recent study shows that aggregates of medin have been found in the brain vasculature of wild-type mice in an age-dependent manner and are closely associated with age-related cognitive decline [1]. Strikingly, genetic deficiency of the medin precursor protein, MFG-E8, eliminates not only vascular aggregates but also prevents age-associated decline of cerebrovascular function in mice [1].

Remarkably, cerebrovascular medin also is a strong predictor of AD [4], and a recent study demonstrates that medin peptide forms amyloids in cerebrovascular walls and promotes vascular Aβ deposits by fibrilizing them with Aβ; in contrast, lowering medin protects against cerebral amyloid angiopathy and cognition impairment in mice by a cross-seeding mechanism [27]. Thus, medin could be a biomarker diagnostic tool for AD.

## 7. Concluding Remarks and Perspectives

The medin precursor MFG-E8 is mainly secreted by VSMCs and is markedly increased in the aging arterial wall. Medin is an internal cleavage product from the second C-terminal discoid domain of MFG-E8 and is markedly increased in the aging arterial wall. Medin amyloid is commonly observed in the aged upper arterial beds: aortae, temporal arteries, and intracranial arteries. Both ECs and VSMCs treated with medin release inflammatory cytokines and chemokines. Importantly, treating ECs with medin promotes apoptosis and necrosis and reduces their viability, while treating VSMCs with medin promotes senescence, necrosis and enhances inflammatory secretion. MFG-E8 and medin avidly bind to elastic fibers in a dose-dependent concentration exerting matrix modifications such as amyloidosis, elastolysis, and calcification. From functional views, the vascular basic elastin-contractile unit bound to MFG-E8/ medin becomes stiffened, and the vascular endothelium bound to medin becomes disordered, facilitating the decline of endothelial-dependent relaxation and the stiffening of the arterial wall. The MFG-E8/medin associated cellular and matrix structural and functional adverse remodeling is fertile soil for the development of age-associated arterial diseases such as aneurysm/dissection, temporal arteritis, and cerebrovascular cognition decline or dementia.

Targeting MFG-E8/medin is a novel approach to diagnosing and managing arterial aging and the age-associated cardiovascular and cerebrovascular disease: (1) an inhibition of MFG-E8 expression and signaling could be alleviated in Ang II induced vascular inflammation during aging and hypertension; (2) a mitigation of MFG-E8 and signaling could be a promising approach to preventing or treating arterial stenosis and calcification in angioplasty after injury and aging; (3) the alleviation of MFG-E8 expression and its conversion into medin could be a novel therapeutic approach to the treatment and prevention of temporal arteritis, vascular-related cognitive impairment, and dementia as well as Alzheimer’s disease. Thus, targeting the expression of both MFG-E8 and medin as well as their signaling is a biomarker diagnostic tool and is also a novel and potential molecular approach to slowing aging and age-related cardiovascular and cerebrovascular disease.

## Figures and Tables

**Figure 1 cells-12-00253-f001:**
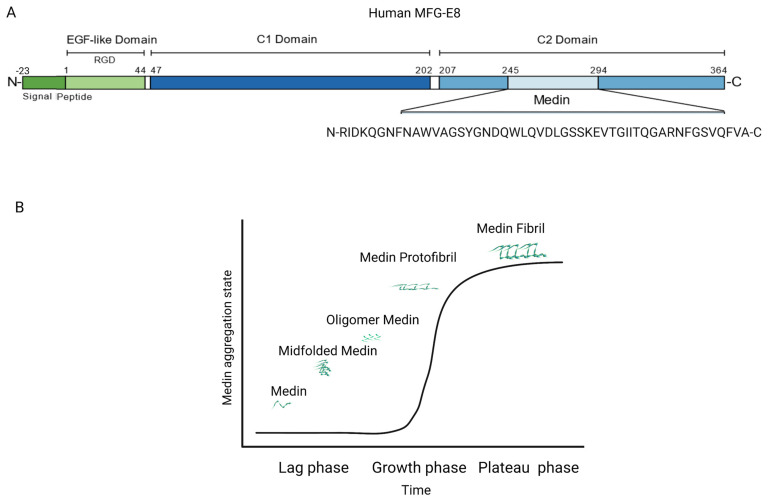
Schematic of MFG-E8, medin, and medin amyloid formation. (**A**) MFG-E8 is a secreted glycoprotein and is also termed lactadherin. Human MFG-E8 contains an EGF-like domain on the N-terminal side and two repeated domains homologous to blood coagulation factor V/VIII (C1 and C2 domains, respectively) on the C-terminal side. Medin is a 50-amino acid peptide cleaved by unknown mechanisms from the C2 domain of MFG-E8. (**B**) An in vitro assay of medin aggregation state in three phases: the lag phase, growth phase, and plateau phase. MFG-E8, milk fat globule EGF VIII; EGF, epidermal growth factor; C1, discoidin-like domain 1; C2, discoidin-like domain 2; RGD, an Arg-Gly-Asp motif. Created with BioRender.com.

**Figure 2 cells-12-00253-f002:**
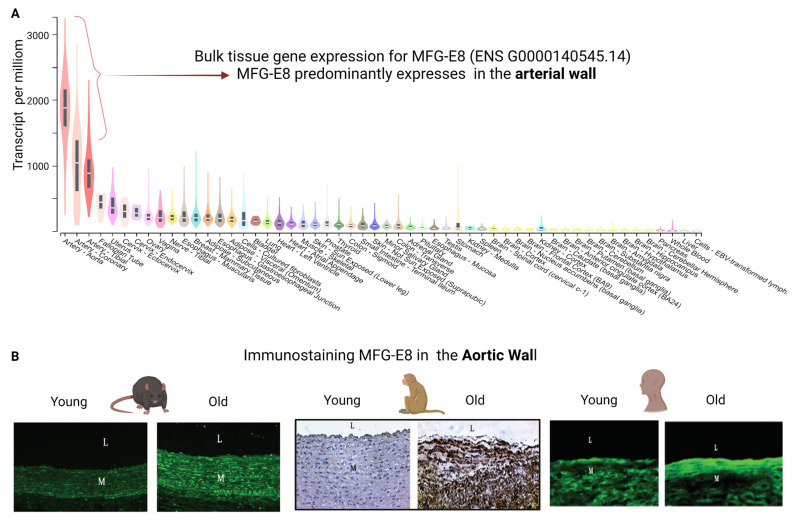
MFG-E8 expression in the aortic wall. (**A**) Bulk human tissue expression for MFG-E8 from the genotype-tissue expression (GTEx) project (version 8) (http://gtexportal.org/omr/ (accessed on 23 October 2022)), showing that MGF-E8 is highly expressed in arteries, especially in the aorta. (**B**) Immunostaining MFG-E8 in the aortic walls in aging rats, nonhuman primates and human samples, modified from Wang M et al. [10]. MFG-E8, milk fat globule EGF VIII. L, lumen; M. media. Created with BioRender.com.

**Figure 3 cells-12-00253-f003:**
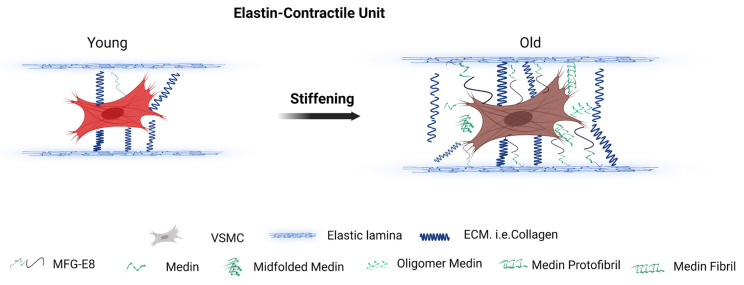
Elastin-contractile unit in the vascular wall. A schematic depiction of remodeling of the arterial elastin-contractile unit with aging, focusing on MFG-E8/medin. MFG-E8/medin acts as a bridging molecule (linker) between the elastic lamina and VSMCs which is involved in the stiffening of the vascular elastin-contractile microstructure with aging. Cleavage of MFG-E8 to release medin will break this bridge/linkage. Medin may compete with MFG-E8 in the binding to elastic fiber forming a linear amyloid fibril or aggregates as a non-amyloid oligomer, which promotes vascular microstructure stiffening in advanced age. ECM, extracellular matrix; VSMCs, vascular smooth muscle cells; MFG-E8, milk fat globule EGF VIII. Created with BioRender.com.

**Figure 4 cells-12-00253-f004:**
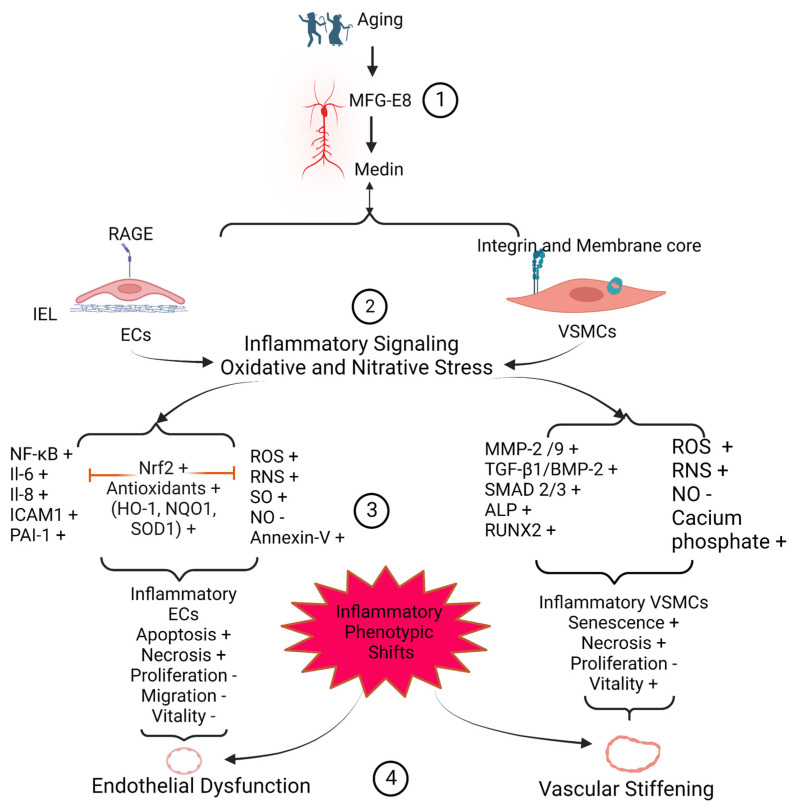
MFG-E8/Medin-mediated vascular cellular inflammation and vascular dysfunction. (1) Aging increases the expression of MFG-E8 and medin; (2) MFG-E8/medin activates the inflammatory reaction and oxidative stress through the RAGE receptor in endothelial cells and triggers the inflammation and oxidation in vascular smooth muscle cell via receptors integrin and oligomer medin forming the membrane pore; (3) MFG-E8/medin associated oxidative and nitrative stress and inflammation eventually leads to the apoptosis, senescence, and necrosis of both ECs and VSMCs; and (4) Vascular inflammatory phenotypic shifts cause vascular endothelial dysfunction (acetylcholine dependent dilatation decline) and vascular stiffening. Abbreviations: MFG-E8, milk fat globule EGF VIII; EC, endothelial cells; IEL, internal elastic lamina; VSMC, vascular smooth muscle cells; RAGE, receptor for advanced glycation end-product; NF-κB, nuclear factor kappa-light-chain-enhancer of activated B cells; IL-6, interleukin-6; IL-8, interleukin-8; ICAM1, intercellular adhesion molecule 1; VCAM1, vascular cellular adhesion molecular 1; PAI 1, plasminogen activator inhibitor 1; ROS, reactive oxygen species; RNS, reactive nitrative species; SO, superoxide; NO, nitric oxide; MMP-2, matrix metalloproteinase type-2; MMP-9, matrix metalloproteinase type-9; TGF-β1, transforming growth factor-beta 1; BMP-2, bone morphogenetic protein-2; SMAD-2/3, suppressor of mothers against decapentaplegic-2/3; ALP, alkaline phosphatase; Runx2, runt-related transcription factor-2; HO-1, heme oxygenase; NQO1, NAD(P)H quinone dehydrogenase-1; Nrf2, nuclear factor erythroid 2-related factor. Created with BioRender.com.

**Figure 5 cells-12-00253-f005:**
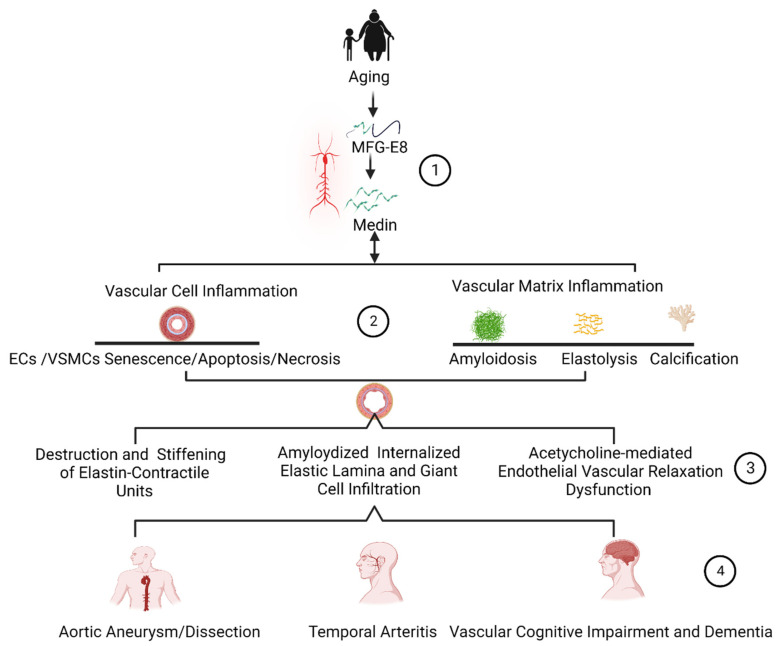
The Role of MFG-E8 Fragment Medin in Age-Associated Arterial Disease. (1) Aging increases MFG-E8 and its fragment medin during inflammatory remodeling; (2) Medin drives the inflammatory phenotypic shifts of ECs and VSMCs, and medin inflammation modifies the vascular extracellular matrix such as through amyloidosis, elastic fragmentation, and calcification; (3) Medin inflammatory signaling promotes vascular wall weakness, endothelial-dependent relaxation decline and arterial stiffening; (4) Medin is involved in the development of vascular diseases, aortic aneurysm/dissection, arteritis, and vascular dementia. MFG-E8, milk fat globule EGF VIII; EC, endothelial cells; VSMCs, vascular smooth muscle cells. Created with BioRender.com.

**Table 1 cells-12-00253-t001:** Medin amyloid fibril formation in vitro.

Amyloid Seed	Congo Red Staining	Electron Microscopy	Time to Form Amyloid Fibrils
Synthetic peptide			
Medin-aa _1–50_	+	+	5–7 days
Medin-aa _11–50_	+	+	8–10 days
Medin-aa _31–50_	+	+	1 day
Medin-aa _21–41_	+	+	Instantly
Medin-aa _42–49_	+	+	2–4 days
Medin-aa _42–50_	+	+	Instantly
Recombinant peptide Medin-aa _1–50_	+	+	Instantly

Congo red staining +, medin or medin fragment amyloid fibrils appear with a typical apple-green birefringence color under the polarized microscope; Electron microscopy +, aggregated medin or medin fragments show fibril formation under the electron microscope.

## Data Availability

Not applicable.

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
