# Peer review of "Milk Fat Globule Epidermal Growth Factor VIII Fragment Medin in Age-Associated Arterial Adverse Remodeling and Arterial Disease"

_cells, 2023, doi:10.3390/cells12020253_

Round 1

Reviewer 1 Report

The aim of the review is to summarize the effects of the peptide Medin and its parent molecule MFG-E8 on vascular aging. The review starts by describing the inflammatory effects induced by MGF-E8 and Medin, both in vascular endothelial cells and in smooth muscle. Thereafter the contribution of Medin/MFG- E8 to amyloid deposits between smooth muscle cells in elastic arteries and its co-localization with amyloid β is described. Further, the involvement of medin in the degeneration and association to elastic fibres as well as in promoting calcification of arteries and contributing to endothelial/vascular dysfunction is discussed. Finally, Medins involvement in the pathogeneis of aneurysms, arteritis and dementia is further described.

The current review summarizes an interesting and important topic and overall it is well written. It is however, not clearly stated what knowledge gap that needs to be fulfilled and how MFGE8/Medin can be targeted for future interventions. The aim of the review should be clearly stated.

The manuscript need some modifications to aid in enhancing the understanding for the reader. For example, the exact functions of Medin and/or MFG-E8 is hard to grasp and a more comprehensive explanation of the mechanisms related to Medin and those related to MFGE8 would be helpful.

The first part of the paper regarding MFG-E8 and Medins role in inflammation has been reviewed in 2020 by Ni et al in Ageing Research Rewiews (Roles and mechanisms of MFG-E8 in vascular aging related diseases). This review is not referred to in the current review. The authors should refer to this review and perhaps shorten the sections on inflammation. Also, there is no mention in the current review of the fact that MFGE8 actually increases proliferation which is a bit counterintuitive regarding development of aneurysms. This should also be discussed. The current review, however, contains additional aspects of the pathogenesis of vascular diseases, not covered by Ni et al.

In general, the references cited are appropriate for the content of the review.

The paper only contains one figure, and adding more figures would enhance the understanding. When looking at MFG-E8 in the GTeX database (https://www.gtexportal.org/home), it is very clear that MGFE8 is highly expressed in arteries, especially aorta (see figure 1 below). The data from GTex could be used as an illustrative picture.

Figure 1.

A figure on how MFGE8/ medin is involved in the contractile-elastic unit would be very helpful. Also, the references regarding the contractile elastic unit on page 1 row 40, do not properly explain this as far as I understand.

It is not clear exactly how medin/MFGE8 causes vascular endothelial dysfunction. On page 6 row 269-271, there is no reference regarding this sentence.

On page 5 row 171-172, there is an explanation on how different Medin amyloid fibrils form. This is hard to read and a table would be suitable for this data.

In the end of page 8, row 372-374, there is a description of how Medin could be bridging of cells to other cells or surfaces. It is not clear how this is accomplished and a figure would help.

Even though the paper in general is well written, it still needs some editing, because some sentences seem to lack words and other seem to have too many. The text should be checked carefully.

Author Response

Reviewer 1: Comments and Suggestions for Authors
The aim of the review is to summarize the effects of the peptide Medin and its parent molecule
MFG-E8 on vascular aging. The review starts by describing the inflammatory effects induced by
MGF-E8 and Medin, both in vascular endothelial cells and in smooth muscle. Thereafter the
contribution of Medin/MFG-E8 to amyloid deposits between smooth muscle cells in elastic
arteries and its co-localization with amyloid β is described. Further, the involvement of medin in
the degeneration and association to elastic fibres as well as in promoting calcification of arteries
and contributing to endothelial/vascular dysfunction is discussed. Finally, Medins involvement in
the pathogenesis of aneurysms, arteritis and dementia is further described.

Thanks for reviewing our manuscript.

The current review summarizes an interesting and important topic and overall, it is well written. It
is however, not clearly stated what knowledge gap that needs to be fulfilled and how
MFGE8/Medin can be targeted for future interventions. The aim of the review should be clearly
stated.

Thanks for your constructive suggestion. In the revised version, we have pointed out the
knowledge gap concerning MFG-E8/medin, for example, it is unknown how MFG-E8 is
cleaved into its fragment medin in the arterial with advancing age. We have made some
statements that targeting the turnover of MFGE8 to its fragment medin and their signalling
could be a novel approach to future interventions of arterial aging and age-related
cardiovascular disease in the section of concluding remarks and perspectives.

The manuscript needs some modifications to aid in enhancing the understanding for the reader.
For example, the exact functions of Medin and/or MFG-E8 is hard to grasp, and a
more
comprehensive explanation of the mechanisms related to Medin and those related to MFGE8
would be helpful.

In the revised version, we have substantially revised relevant contents and illustrated the
structure and function of MFG-E8 and its fragment medin in the vascular system (Figures
1-5). We believe these figures in the revised version are helpful to the further understanding
of the inflammatory role of MFG-E8 fragment medin during adverse vascular remodelling
and disease with advancing age.

The first part of the paper regarding MFG-E8 and Medins role in inflammation has been reviewed
in 2020 by Ni et al in-Ageing Research Rewiews (Roles and mechanisms of MFG-E8 in vascular
aging related diseases). This review is not referred to in the current review. The authors should
refer to this review and perhaps shorten the sections on inflammation.

In the revised version, we have referred to the review paper: Roles and mechanisms of MFG-
E8 in vascular aging related diseases [1]. In the current review, we have comprehensively
reviewed the inflammatory role of MFG-E8 fragment medin in vascular remodelling and
disease. In addition, we also briefly update a new understanding of the inflammatory role of
MFG-E8 mediated angiotensin II induced arterial remodelling in mice with advancing age
[2].

Also, there is no mention in the current review of the fact that MFGE8 increases proliferation
which is a bit counterintuitive regarding development of aneurysms. This should also be discussed.

MFG-E8 signaling, indeed, is closely associated with the proliferation of both VSMCs in vitro
and in the vascular wall in mice or rats [2,3]. Notably, MFG-E8 fragment medin is
significantly or predominantly increased in the local vascular area during the development
of arterial aneurysm/dissection[4,5]. In the revised version, we have discussed that the
rationale of
MFGE8 being involved in the development of aneurysms is mainly associated
with its degradative product medin peptide. The medin or medin oligomer increases the
activation of MMP-2 and the necrosis of VSMCs as well as the stiffness of vascular
microstructure, which is a potential molecular mechanism behind the MFG-E8 associated
development of aneurysm.

In general, the references cited are appropriate for the content of the review.

Thanks for commenting about the references.

The paper only contains one figure and adding more figures would enhance the understanding.
When looking at MFG-E8 in the GTeX database (https://www.gtexportal.org/home), it is very
clear that MGFE8 is highly expressed in arteries, especially aorta (see figure 1 below). The data
from GTex could be used as an illustrative picture.

Thank you for your thoughtful suggestions. The MFG-E8 data from GTex have be used as
an illustrative picture in the revised Figure 1A. In addition, we have added the
immunostaining of MFG-E8 in the mouse, rat, monkey, and human aortic samples
enhancing/contributing to the further understanding of the role of MFG-E8 in age-related
inflammatory aortic remodelling in the revised Figure 1B, modified from Wang M et al [6].

A figure on how MFGE8/ medin is involved in the contractile-elastic unit would be very helpful.
Also, the references regarding the contractile elastic unit on page 1 row 40, do not properly explain
this as far as I understand.

Thanks for your suggestion. The MFG-E8/medin-related elastin-contractile unit/stiffening
has been illustrated in the revised Figure 2.

It is not clear exactly how medin/MFGE8 causes vascular endothelial dysfunction. On page 6 row
269-271, there is no reference regarding this sentence.

In the revised version, we have clarified how medin/MFGE8 causes vascular endothelial
dysfunction and the revised figure 4 illustrates the potential mechanisms behind these effects.
In addition, we have modified the statement the original version page 6 row 269-271) and
cited the related reference [7,8].

On page 5 row 171-172, there is an explanation on how different Medin amyloid fibrils form. This
is hard to read, and a table would be suitable for this data.

Thanks for your suggestion. In the revised version, Table 1 shows medin amyloid fibrils
formation by different medin/medin fragment seeds in vitro.

In the end of page 8, row 372-374, there is a description of how Medin could be bridging of cells
to other cells or surfaces. It is not clear how this is accomplished, and a figure would help.

In the revised Figure 2 shows that medin could acts as a liner/bridging molecule between the
surface of VSMCs and elastic laminae in the elastin-contractile unit.

Even though the paper in general is well written, it still needs some editing, because some
sentences seem to lack words and other seem to have too many. The text should be checked
carefully.

We have edited the manuscript carefully in the revised version.

Responding references

1. Ni, Y.Q.; Zhan, J.K.; Liu, Y.S. Roles and mechanisms of MFG-E8 in vascular aging-
related diseases. Ageing Res Rev 2020, 64, 101176, doi:10.1016/j.arr.2020.101176.

2. Ni, L.; Liu, L.; Zhu, W.; Telljohann, R.; Zhang, J.; Monticone, R.E.; McGraw, K.R.; Liu,
C.; Morrell, C.H.; Garrido-Gil, P.; et al. Inflammatory Role of Milk Fat Globule-Epidermal
Growth Factor VIII in Age-Associated Arterial Remodeling. J Am Heart Assoc 2022, 11,
e022574, doi:10.1161/jaha.121.022574.

3. Wang, M.; Fu, Z.; Wu, J.; Zhang, J.; Jiang, L.; Khazan, B.; Telljohann, R.; Zhao, M.; Krug,
A.W.; Pikilidou, M. MFG‐E8 activates proliferation of vascular smooth muscle cells via
integrin signaling. Aging cell 2012, 11, 500-508.

4. Davies, H.A.; Caamano-Gutierrez, E.; Chim, Y.H.; Field, M.; Nawaytou, O.; Ressel, L.;
Akhtar, R.; Madine, J. Idiopathic degenerative thoracic aneurysms are associated with
increased aortic medial amyloid. Amyloid 2019, 26, 148-155,
doi:10.1080/13506129.2019.1625323.

5. Peng, S.; Larsson, A.; Wassberg, E.; Gerwins, P.; Thelin, S.; Fu, X.; Westermark, P. Role
of aggregated medin in the pathogenesis of thoracic aortic aneurysm and dissection. Lab
Invest 2007, 87, 1195-1205, doi:10.1038/labinvest.3700679.

6. Wang, M.; H Wang, H.; G Lakatta, E. Milk fat globule epidermal growth factor VIII
signaling in arterial wall remodeling. Current vascular pharmacology 2013, 11, 768-776.

7. Karamanova, N.; Truran, S.; Serrano, G.E.; Beach, T.G.; Madine, J.; Weissig, V.; Davies,
H.A.; Veldhuizen, J.; Nikkhah, M.; Hansen, M.; et al. Endothelial Immune Activation by
Medin: Potential Role in Cerebrovascular Disease and Reversal by Monosialoganglioside-
Containing Nanoliposomes. J Am Heart Assoc 2020, 9, e014810,
doi:10.1161/JAHA.119.014810.

8. Migrino, R.Q.; Davies, H.A.; Truran, S.; Karamanova, N.; Franco, D.A.; Beach, T.G.;
Serrano, G.E.; Truong, D.; Nikkhah, M.; Madine, J. Amyloidogenic medin induces
endothelial dysfunction and vascular inflammation through the receptor for advanced
glycation endproducts. Cardiovasc Res 2017, 113, 1389-1402, doi:10.1093/cvr/cvx135.

Reviewer 2 Report

This is a review article on medin,  the most common amylogenic protein found in arterial wall of ageing arteries, but also in vascular remodeling and inflammation.

The flow of the article is very nice; however, I would suggest to provide more details from original studies in some points for readers that are not fully familiar with medin.

line 98 "Medin-associated oxidative stress and 98 inflammatory reactions induces vascular cell inflammatory phenotypic shifts..."- could you please further elaborate in the text original findings from quoted references (1,2,4,16,22,24,31).

Please indicate models when is possible (e.g. human arterial tissue, cell or tissue cultures, rodent experimental model etc).

line-70-72" angiotensin II (Ang II)/ its  receptor AT1 protein signaling and nuclear transcriptional factor-kappa B (NF-κB) transcriptional activation; in contrast, MFG-E8 deficiency markedly alleviates these age effects  [26,27]." These were studies with infusion of ANG II. WHat was the protocol? What was the does and duration of infusion of ANG II? Was blood pressure increased in these studies? How would you comment potential long-term increases in blood pressure, such as in hypertension and medin expression and role in vascular remodeling?

line 291 and that paragraph- please elaborate on "Accumulating evidence indicates that medin is involved...." - include examples and more details on direct involvement of medin in the mechanisms of endothelium -dependent vascular function, production of ROS etc...[2,16,24,36,43].

395.-398 "Old medin amyloid is a  key element of the neurovascular unit in the brain, causing microvascular endothelial dys- function through oxidative and nitrative stress and promotes pro-inflammatory signaling n EC via RAGE showing EC immune activation and neuroinflammation in the elderly  [2,9]." Could you elaborate on this- are there direct effects or it is just speculation on the mechanisms? Could you make some kind of schematic?

Are there any experiments beside quoted one at the end of the manucriot with e.g. knock out mice or rats or cells that show restoration of dysfuctional phenotype when medin is excluded? It seems tbat medin in normal component of arterial wall in ageing. Are there any evidence of different expression or function of medin in diseases, compared to normal vascular ageing?

Author Response

Reviewer 2: Comments and Suggestions for Authors
This is a review article on medin, the most common amylogenic protein found in arterial wall of
ageing arteries, but also in vascular remodeling and inflammation.

The flow of the article is very nice; however, I would suggest providing more details from original
studies in some points for readers that are not fully familiar with medin.

line 98 "Medin-associated oxidative stress and 98 inflammatory reactions induces vascular cell
inflammatory phenotypic shifts..."- could you please further elaborate in the text original findings
from quoted references (1,2,4,16,22,24,31). Please indicate models when is possible (e.g., human
arterial tissue, cell or tissue cultures, rodent experimental model etc.).

Thanks for your thoughtful comments. Medin associated oxidative stress and inflammation
induces vascular cell inflammatory phenotypic shifts that have been elaborated, clarified,
and illustrated in the revised version and Figure 3.

line-70-72" angiotensin II (Ang II)/ its receptor AT1 protein signaling and nuclear transcriptional
factor-kappa B (NF-κB) transcriptional activation; in contrast, MFG-E8 deficiency markedly
alleviates these age effects [26,27]." These were studies with infusion of ANG II. What was the
protocol? What was the does and duration of infusion of ANG II? Was blood pressure increased
in these studies? How would you comment potential long-term increases in blood pressure, such
as in hypertension and medin expression and role in vascular remodeling?

In the revised, the relevant issues/questions have been addressed. long-term increases in age-
associated blood pressure increase, MFG-E8 and its fragment medin expression increases

and exert the inflammatory role in vascular structural and function remodeling such as
intimal-medial thickening, endothelial-dependent relaxation, cognition impairment and
dementia [1,2].

line 291 and that paragraph- please elaborate on "Accumulating evidence indicates that medin is
involved...." - include examples and more details on direct involvement of medin in the
mechanisms of endothelium-dependent vascular function, production of ROS etc. [2,16,24,36,43].

The relevant comments have been elaborated in the revised version and illustrated in Figure
3.

395.-398 "Old medin amyloid is a key element of the neurovascular unit in the brain, causing
microvascular endothelial dysfunction through oxidative and nitrative stress and promotes pro-
inflammatory signaling in EC via RAGE showing EC immune activation and neuroinflammation
in the elderly [2,9]." Could you elaborate on this- are there direct effects or it is just speculation
on the mechanisms? Could you make some kind of schematic?

The relevant statements have been clarified and schematically illustrated in the revised
Figure 3.

Are there any experiments beside quoted one at the end of the manuscript with e.g., knock out
mice or rats or cells that show restoration of dysfunctional phenotype when medin is excluded? It
seems that medin in normal component of arterial wall in ageing. Are there any evidence of
different expression or function of medin in diseases, compared to normal vascular ageing?

MFG-E8 knockout mice that show restoration of inflammation, vascular dysfunctional and
cognition decline resulting from medin co-arrogates with vascular amyloid when medin is
excluded [1,3,4]

Are there any evidence of different expression or function of medin in diseases, compared to
normal vascular ageing?

The study shows that knockout MFG-E8 of rat VSMCs restores MFG-E8 associated with
the inactivation of MMP-2 and the cleavage of tropoelastin [5]. Silencing human HUVEC
MFG-E8 substantially reduced VSMC senescence and calcification and abolished the
osteogenic protein ALP2. Runx2 expression in human VSMCs induced by HG-induced
HVEC-released exosomes [6] In addition, MFG-E8 and MFG-E8 fragment medin levels were
significantly expressed in arterial stenosis, aneurysm, and cognitive decline) compared to
normal vascular ageing [7-11].

Responding references

1. Ni, L.; Liu, L.; Zhu, W.; Telljohann, R.; Zhang, J.; Monticone, R.E.; McGraw, K.R.; Liu,
C.; Morrell, C.H.; Garrido-Gil, P.; et al. Inflammatory Role of Milk Fat Globule-Epidermal
Growth Factor VIII in Age-Associated Arterial Remodeling. J Am Heart Assoc 2022, 11,
e022574, doi:10.1161/JAHA.121.022574.

2. Wang, M.Y.; Telljohann, R.; Ni, L.; Zhu, W.Q.; Kim, S.H.; Liu, L.J.; Zhang, J.; McGraw,
K.; Monticone, R.; Morrell, C.; et al. Effects of Milk Fat Globule Epidermal Growth Factor
VIII on Adverse Aortic Remodeling in Mice With Advancing Age. Circulation 2021, 144,
A9460-A9460.

3. Degenhardt, K.; Wagner, J.; Skodras, A.; Candlish, M.; Koppelmann, A.J.; Wild, K.;
Maxwell, R.; Rotermund, C.; von Zweydorf, F.; Gloeckner, C.J.; et al. Medin aggregation
causes cerebrovascular dysfunction in aging wild-type mice. Proc Natl Acad Sci U S A
2020, 117, 23925-23931, doi:10.1073/pnas.2011133117.

4. Wagner, J.; Degenhardt, K.; Veit, M.; Louros, N.; Konstantoulea, K.; Skodras, A.; Wild,
K.; Liu, P.; Obermüller, U.; Bansal, V.; et al. Medin co-aggregates with vascular amyloid-
β in Alzheimer's disease. Nature 2022, doi:10.1038/s41586-022-05440-3.

5. Kim, S.H.; Liu, L.; Ni, L.; Zhang, L.; Zhang, J.; Wang, Y.; McGraw, K.R.; Monticone, R.;
Telljohann, R.; Morrell, C.H.; et al. Effects of Milk Fat Globule Epidermal Growth Factor
VIII On Age-Associated Arterial Elastolysis, Fibrosis, and Calcification. bioRxiv 2022,
2020.2010.2005.326728, doi:10.1101/2020.10.05.326728.

6. Ni, Y.-Q.; Li, S.; Lin, X.; Wang, Y.-J.; He, J.-Y.; Song, W.-L.; Xiang, Q.-Y.; Zhao, Y.; Li,
C.; Wang, Y.; et al. Exosomal MFGE8 from high glucose induced endothelial cells is
involved in calcification/senescence of vascular smooth muscle cells. bioRxiv 2021,
2021.2009.2010.459867, doi:10.1101/2021.09.10.459867.

7. Chiang, H.Y.; Chu, P.H.; Lee, T.H. MFG-E8 mediates arterial aging by promoting the
proinflammatory phenotype of vascular smooth muscle cells. J Biomed Sci 2019, 26, 61,
doi:10.1186/s12929-019-0559-0.

8. Peng, S.; Larsson, A.; Wassberg, E.; Gerwins, P.; Thelin, S.; Fu, X.; Westermark, P. Role
of aggregated medin in the pathogenesis of thoracic aortic aneurysm and dissection. Lab
Invest 2007, 87, 1195-1205, doi:10.1038/labinvest.3700679.

9. Davies, H.A.; Caamano-Gutierrez, E.; Chim, Y.H.; Field, M.; Nawaytou, O.; Ressel, L.;
Akhtar, R.; Madine, J. Idiopathic degenerative thoracic aneurysms are associated with

increased aortic medial amyloid. Amyloid 2019, 26, 148-155,
doi:10.1080/13506129.2019.1625323.

10. Migrino, R.Q.; Karamanova, N.; Truran, S.; Serrano, G.E.; Davies, H.A.; Madine, J.;
Beach, T.G. Cerebrovascular medin is associated with Alzheimer's disease and vascular
dementia. Alzheimers Dement (Amst) 2020, 12, e12072, doi:10.1002/dad2.12072.

11. Migrino, R.Q.; Davies, H.A.; Truran, S.; Karamanova, N.; Franco, D.A.; Beach, T.G.;
Serrano, G.E.; Truong, D.; Nikkhah, M.; Madine, J. Amyloidogenic medin induces
endothelial dysfunction and vascular inflammation through the receptor for advanced
glycation endproducts. Cardiovasc Res 2017, 113, 1389-1402, doi:10.1093/cvr/cvx135.

Reviewer 3 Report

There are already reviews with a similar topic available.

The title does not fit the content perfectly. Are all described effects of medin inflammatory or could there also be inflammation induced “medin production”? See also fig 1 upper part e.g. vascular cell inflammation -> Medin.  A somewhat broader title would be more appropriate.

What is cause and effect? In line 87, page 3: “to elucidate the underlying molecular mechanisms behind the conversion/turnover of the MFG-E8-Medin in the arterial wall”. The authors’ statement should be adjusted. It is not clear what comes first. On the one hand oxidative stress induces medin (line 91), on the other hand “…medin-associated oxidative stress induces…”(line 98)  The review describes more the effects of MFG-E8/medin, whereas the molecular mechanism of conversion is not so clear. See also page 4, line 134-138.

General references and MFG-E8/medin related references (most citations are from 2-3 groups) are not balanced. Reduction of references to the important ones would help to read the review. 

I am not sure whether citation of bioRxiv manuscripts is possible/allowed.

Typos:

Page 5, line 212 ”…in an along..”  -> “.. in and along…”

Page 6, line 240 ”…MFG-E8 knock mice...”  -> “.. MFG-E8 knockout mice …”

Author Response

Response to reviewer 3’s comments and suggestions for authors
There are already reviews with a similar topic available.

The previous two review articles mainly focus on the effect of MFG-E8 on vascular remodeling
and age-related disease [1,2]. In the current review, we mainly focus on the inflammatory MFG-
E8 fragment medin in the vascular system and vascular disease and briefly update on the effect
of MFG-E8 on angiotensin II mediated vascular remodeling. Therefore, this manuscript
specifically reviews and updates on the role of MFG-E8 fragment medin in the vascular cell,
vascular tissue, vascular related diseases with advancing age.

The title does not fit the content perfectly. Are all described effects of medin inflammatory or
could there also be inflammation induced “medin production”? See also fig 1 upper part e.g.,
vascular cell inflammation -> Medin. A somewhat broader title would be more appropriate.

We admit the current title is imperfect, but we capture the main review theme that MFG-E8
fragment medin, oligomer medin, or amyloid medin signaling exerts an inflammatory role in
vascular cells, vascular wall, vascular diseases such as aneurysm/dissection, giant cell arteritis,
and vascular related cognitive decline and Alzheimer dementia. In the revised version, this
manuscript has been entitled “MFG-E8 fragment medin in age-associated arterial adverse
remodeling and arterial disease”. In addition, we have updated recent advances of MFG-E8 in
vascular aging.

What is cause and effect? In line 87, page 3: “to elucidate the underlying molecular mechanisms
behind the conversion/turnover of the MFG-E8-Medin in the arterial wall”. The authors’ statement
should be adjusted. It is not clear what comes first. On the one hand oxidative stress induces medin

(line 91), on the other hand “...medin-associated oxidative stress induces...” (line 98) The review
describes more the effects of MFG-E8/medin, whereas the molecular mechanism of conversion is
not so clear. See also page 4, line 134-138.

In the revised version, the relevant statements have been further clarified. Notably, MFG-E8 is
the medin parent molecule. It is unknown how medin is enzymatically cleaved from the second
discoidin domain from the MFG-E8 molecules so far, which warrants further investigation
(Figure 1 in the revised version).

General references and MFG-E8/medin related references (most citations are from 2-3 groups) are
not balanced. Reduction of references to the important ones would help to read the review.

In the revised version, MFG-E8/medin associated medin references have been updated for more
balance and the general references have been reduced for increasing readability.

I am not sure whether citation of bioRxiv manuscripts is possible/allowed.

As open access policy to scientific research continues to widen, journals, perhaps including
Cells, have eased their policies regarding preprints, now allowing authors to submit manuscripts
that have been posted as preprints and allowing preprints to be cited in reference lists of
submitted articles.

Typos:

Page 5, line 212” ...in an along. -> “.. in and along...”

Page 6, line 240” ...MFG-E8 knock mice...” -> “.. MFG-E8 knockout mice ...”
In the revised version, the type errors have been corrected.

Responding references

1. Ni, Y.Q.; Zhan, J.K.; Liu, Y.S. Roles and mechanisms of MFG-E8 in vascular aging-
related diseases. Ageing Res Rev 2020, 64, 101176, doi:10.1016/j.arr.2020.101176.

2. Wang, M.; H Wang, H.; G Lakatta, E. Milk fat globule epidermal growth factor VIII
signaling in arterial wall remodeling. Current vascular pharmacology 2013, 11, 768-776.

Round 2

Reviewer 1 Report

The authors have revised the manuscript accordingly and added several figures and I think that it is significantly improved.

I only have a few very minor comments:

Row 30: This part of the sentence needs to be revised: "Age-associated adverse vascular adversely restructuring, "

Row 78: "Our current review mainly focuses on 1) ...." I think you need to add "how" before the listings.

Author Response

Response to reviewer 1 comments and suggestions
The authors have revised the manuscript accordingly and added several figures and I think that it is
significantly improved.

I only have a few very minor comments:

Row 30: This part of the sentence needs to be revised: "Age-associated adverse vascular adversely
restructuring, "

Thanks for your comment. In the revised version, this statement has been revised as “age-associated
adverse vascular remodeling”

Row 78: "Our current review mainly focuses on 1) ...." I think you need to add "how" before the

In the revised version, we have modified this sentence.

Reviewer 2 Report

Thanks authors for revisions. I have no further comments.

Reviewer 3 Report

no

Round 3

Reviewer 1 Report

I think that the manuscript now is acceptable for publication.